**Data Availability Statement:** Data cannot be shared publicly because of the ethics committee

# Changes in motor paralysis involving upper extremities of outpatient chronic stroke patients from temporary rehabilitation interruption due to spread of COVID-19 infection: An observational study on pre- and post-survey data without a control group

**Daigo Sakamoto**[1,2], **Toyohiro Hamaguchi**[2]*, **Yasuhide Nakayama**[1,3], **Takuya Hada**[3], **Masahiro Abo**[3]*

**1** Department of Rehabilitation Medicine, The Jikei University School of Medicine Hospital, Tokyo, Japan, **2** Department of Rehabilitation, Graduate School of Health Science, Saitama Prefectural University, Saitama, Japan, **3** Department of Rehabilitation Medicine, The Jikei University School of Medicine, Tokyo, Japan

* hamaguchi-toyohiro@spu.ac.jp (TH); abo@jikei.ac.jp (MA)

## Abstract

### Background

Outpatient rehabilitation was temporarily suspended because of coronavirus disease (COVID-19), and there was a risk that patients' activities of daily living (ADLs) would decrease and physical functions unmaintained. Therefore, we investigated the ADLs and motor functions of chronic stroke patients whose outpatient rehabilitation was temporarily interrupted.

### Methods

In this observational study, the Fugl-Meyer Assessment of the Upper Extremity (FMA-UE), Action Research Arm Test (ARAT), and Barthel Index (BI) scores of 49 stroke hemiplegic patients at 6 and 3 months before rehabilitation interruptions were retrospectively determined and were prospectively investigated on resumption of outpatient rehabilitation. Presence or absence of symptoms and difficulties caused by the interruption period (IP) was investigated using a binomial method. Deltas were analyzed using a generalized linear model (GLM) according to the survey period. Age, sex, severity of FMA-UE immediately post-resumption and post-onset period were used as covariates. For survey items showing significant model fit, the 95% confidence interval of minimum detectable change (MDC$_{95}$) was calculated, and the amount of change was compared. Questionnaire responses were tested via proportion ratio. Statistical significance was set at 5%.

### Results

The FMA-UE part A and total scores were significantly model fit depending on periods. The estimated FMA-UE total score decreased by 1.64 (z = −2.38, p = 0.02) during the 3-month

Jikei university. Data are available from Clinical Research Support Center, Jikei University School of Medicine (crb@jikei.ac.jp) for researchers who meet the criteria for access to confidential data." <contact information> Clinical Research Support Center, Jikei University School of Medicine 3-25-8 Nishi-Shimbashi, Minato-ku, Tokyo, 105-8461 Tel: +81-3-3433-1111 Ext 2187 Fax: +81-3-5400-1388 Mail: crb@jikei.ac.jp.

**Funding:** This work was supported by JSPS KAKENHI Grant Number 18K10691.

**Competing interests:** The authors have declared that no competing interests exist.

IP. No fits were observed by GLM in other parts of the FMA-UE, ARAT, or BI. The calculated $MDC_{95}$ was 3.58 for FMA-UE part A and 4.50 for FMA-UE overall. Answers to questions regarding sleep disturbance and physical pain were significantly biased toward "no" in the psychosomatic function items ($p<0.05$). There was no bias in the distribution of answers to questions regarding joint stiffness, muscle weakness, muscle stiffness, and difficulty in moving arms and hands. All 16 questions regarding activities and participation items were significantly biased toward answers "no" ($p<0.05$).

## Conclusions

The FMA-UE part A and total scores were affected. Patients complained of subjective symptoms related to upper limb paralysis after the IP. Since ADLs of patients were maintained, the therapist can recommend that patients not receiving outpatient treatments be evaluated in relation to the shoulder, elbow, and forearm and instructed on self-training to maintain motor function.

## Introduction

Coronavirus disease (COVID-19), a new type of coronavirus infection caused by severe acute respiratory syndrome coronavirus 2 (SARS-CoV-2), was declared a pandemic in March 2020 after being identified in December 2019 [1, 2]. Avoiding close contact with infected persons was recommended [3] because SARS-CoV-2 is transmitted mainly by direct contact, droplets, and airborne transmission [4, 5]. Therefore, lockdown was implemented to restrict people from going out [6, 7] and prevent the spread of SARS-CoV-2 infection. In Japan, the first state of emergency for coronavirus infection was issued on April 7, 2020, which restricted unnecessary and non-urgent travels both within and out of the country [8]. As there are many opportunities for physical contact between patients and therapists, therapists with COVID-19 or their infected close contacts could be a risk factor for cluster infections that transmit the infection to patients [9]. Rehabilitation treatment for outpatients in the chronic phase was discontinued to prevent the influx of viruses from outside the hospital [10].

Lockdowns aimed at preventing the spread of infection have forced people to spend more time at home and change their lifestyles. There was concern that the amount of physical activity would decrease as an adverse effect of staying at home [11]. The results of an online survey conducted by 35 research institutes worldwide reported that spending time at home extended sitting time and reduced physical activity [12]. It was reported that this decrease in physical activity hastened onset of sedentary lifestyle-related diseases and worsened mental illness [13, 14].

Continuous rehabilitation is performed to maintain or improve various functions and abilities of chronic stroke patients with motor paralysis [15, 16]. Chronic stroke patients who require outpatient treatment at the hospital were unable to receive treatment or suggestions from a therapist due to interruptions of outpatient rehabilitation caused by the spread of SARS-CoV-2 infection. Therefore, patients may not be able to maintain their activity of daily living (ADL) functions and physical functions, and may have decreased amounts of activity due to self-restraint from going out. In these cases, when outpatient rehabilitation is resumed, the patient's functional changes should be properly evaluated and appropriate treatment and suggestions should be provided. However, the effects of temporary interruption in outpatient

rehabilitation and prolonged home stay due to the spread of SARS-CoV-2 infection on ADL and physical functions of patients with chronic stroke have been unknown until now. Such information can be used as a reference item that should be evaluated on a regular basis during medical care when an unknown source of infection spreads or when the activity of a patient is restricted due to an unexpected accident. This information will be useful for the development of new rehabilitation medicine techniques using remote communication technology for patients who refrain from going out.

This study aimed to investigate the changes that occurred in outpatients with chronic stroke who had temporary interruption of rehabilitation and refrained from going out by investigating ADL and upper limb motor function during the COVID-19 pandemic.

## Materials and methods

### Participants

The Fugl-Meyer Assessment of Upper Extremity (FMA-UE) was used as the main outcome in this study [17]. Data collection time was set at (1) approximately 6 months before outpatient rehabilitation was resumed, (2) approximately 3 months before outpatient rehabilitation was resumed, and (3) when the outpatient rehabilitation was resumed. Changes in the FMA-UE score during the 3-month period before the interruption of rehabilitation (Δpre-interruption) and during the 3-month period after the resumption of rehabilitation (Δpost-interruption) were compared. Changes in the FMA-UE total score was used to fit the linear model. Sample size was estimated by a priori analysis using G*Power, with $\chi^2$ tests with goodness-of-fit set at $\alpha = 0.05$, $1 - \beta = 0.8$, and effect size of 0.5 [18]. Based on this calculation, the minimum sample size in this study was estimated to be 48.

The eligibility criteria were as follows: those who received outpatient occupational therapy at the Department of Rehabilitation, Jikei University School of Medicine Hospital between June 1, 2019, and May 31, 2020, for ≥3 months; chronic stroke patients older than 20 years; and outpatient rehabilitation was interrupted by the COVID-19 pandemic. Those with a diagnosis of higher brain dysfunction, cognitive dysfunction, and psychiatric disorder were excluded because such conditions may affect the measurements of functional evaluation and understanding and implementations of a questionnaire. Patients who resumed outpatient rehabilitation were asked to join the study after confirming that they met the eligibility criteria and after being provided with written and oral explanations about the study.

### Survey periods and instruments

**Survey periods.** Surveys were conducted as follows: (1) approximately 6 months before rehabilitation interruption (Pre 6 m); (2) approximately 3 months before rehabilitation interruption (Pre IP) (formal interruption was set at April 1, 2020, due to the COVID-19 pandemic); and (3) after the outpatient rehabilitation was resumed (Post IP). The data for the Pre 6 m and Pre IP periods were surveyed retrospectively from medical records using FMA-UE, Action Research Arm Test (ARAT), and Barthel Index (BI) data. Post-IP measurements were also retrospectively conducted. To investigate any functional changes that occurred during interruptions of outpatient rehabilitation, each patient answered a questionnaire once after resuming outpatient rehabilitation (Fig 1).

**Main outcome.** The main outcome was the FMA-UE, a comprehensive assessment battery that tests motor function, balance, sensory function, passive range of motion, and degree of joint pain in post-stroke hemiplegic patients [17]. As for motor function items, voluntary and segregation movements along with the recovery stage of motor paralysis were evaluated. Upper limb items were also extracted and analyzed. Upper limb motor function was scored on

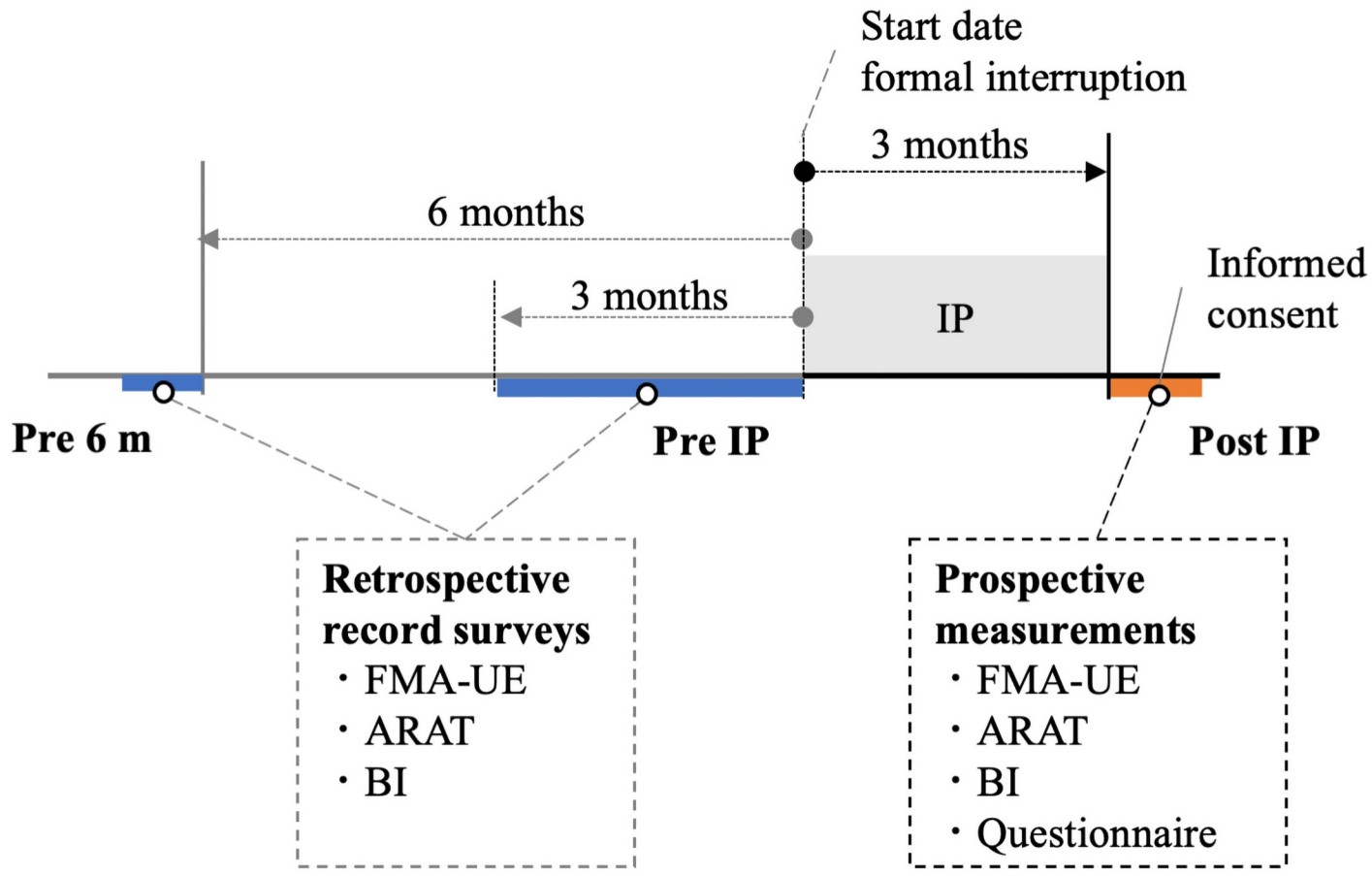

**Fig 1. Survey protocol.** Pre 6 m, approximately 6 months before outpatient rehabilitation was interrupted; Pre IP, approximately 3 months before outpatient rehabilitation was interrupted; IP, interruption period, i.e., during which outpatient rehabilitation was interrupted; Post IP, resumption of outpatient rehabilitation; FMA-UE, Fugl-Meyer Assessment of the Upper Extremity; ARA, Action Research Arm Test; BI, Barthel Index.

a 66-point scale using a three-step ordinal scale. The FMA-UE scores were determined based on the classification reported by Woodbury et al. [19].

**Secondary outcome.** The Action Research Arm Test (ARAT) was used as the secondary outcome [20], and the Barthel Index (BI) [21] was used to evaluate ADL. The ARAT is an upper limb function evaluation tool developed for stroke patients based on the upper extremity test [22]. The ARAT consists of subtests for grasp, grip, pinch, and gross movement and includes tasks on manipulating goods. Each item of the ARAT has a 4-step ordinal scale and is scored with a total of 57 points. The BI was evaluated for the degree of independence in ADL. The BI consists of 10 items: meal, transfer, plastic surgery, excretion, bathing, movement, stair climbing, changing clothes, defecation control, and urination control. All items were scored up to 100 points if patients were completely independent and 0 point if they required assistance at all items. In addition, an original questionnaire was used to investigate subjective changes in mental and physical functions and ADL functions that occurred during the interruption of the outpatient rehabilitation period. The questionnaire was prepared in June 2020 while outpatient rehabilitation was suspended and was devised by occupational therapists working in the rehabilitation department of the Jikei University School of Medicine Hospital. With reference to the International Classification of Functioning, Disability and Health (ICF) category [23], 6 questions related to mental and physical functions and 16 questions related to

activities and participation, for a total of 22 questions, were created (Table 1). Using this questionnaire, the presence or absence of symptoms and subjectively difficult activities of patients during the outpatient rehabilitation IP were investigated using a two-factor method.

**Participant characteristics.** Age, sex, height, weight, and dominant hand were investigated as participant characteristics. Medical information regarding the type of disease (cerebral infarction, cerebral hemorrhage), paralyzed side, post-onset period, IP of outpatient rehabilitation, and whether botulinum treatment was performed was investigated.

**Investigators.** The median values of upper limb motor function and ADL were determined by seven occupational therapists working in a university hospital and engaged in rehabilitation in the area of cerebrovascular disease for more than 5 years.

## Statistical analysis

The total score for FMA-UE and ARAT, the amount of change per sub-item score, and the BI score were calculated using the following equations:

$$\Delta\text{Pre-interruption} = \text{Pre IP} - \text{Pre 6 m score} \tag{1}$$

$$\Delta\text{Post-interruption} = \text{Post IP} - \text{Pre IP score} \tag{2}$$

**Table 1. Questionnaire items.**

| No. | ICF no. | Items |
|-----|---------|-------|
| **Body functions** | | |
| Q1 | b134 | Do you feel that your sleep is getting disturbed? |
| Q2 | b289 | Did you get more pain somewhere in your body? |
| Q3 | b710 | Do you feel that your joints have become difficult to move? |
| Q4 | b730 | Do you feel that your muscles are weakened? |
| Q5 | b735 | Do you feel your muscle tone has increased? |
| Q6 | b760 | Do you find your arms and hands difficult to move? |
| **Activities and participation** | | |
| Q1 | d415 | Do you find it difficult to turn over, get up, and stand up your own? |
| Q2 | d429 | Do you find it difficult to move to a chair or a wheelchair? |
| Q3 | d435 | Do you find it difficult to lift and carry things? |
| Q4 | d445 | Do you find it difficult to use your fine hands and fingers delicately? |
| Q5 | d449 | Do you feel that you use your arms and hands less often? |
| Q6 | d450 | Do you find it harder to walk? |
| Q7 | d520 | Did you find it difficult to take a bath? |
| Q8 | d530 | Did you find it difficult to wash and make up? |
| Q9 | d540 | Did you find it difficult to operate the toilet? |
| Q10 | d550 | Do you find it difficult to change clothes? |
| Q11 | d560 | Do you find it difficult to eat? |
| Q12 | d598 | Did you have less physical care? |
| Q13 | d640 | Did you find it difficult to do household chores? |
| Q14 | d850 | Did you find it harder to work? |
| Q15 | d910 | Did you find it difficult to have the opportunity to connect with the community and society? |
| Q16 | d920 | Do you find it difficult to do hobbies and leisure activities? |

ICF, International Classification of Functioning, Disability and Health.

To test the hypothesis that interruptions of outpatient rehabilitation and refraining from going out associated with the spread of SARS-CoV-2 infection in patients with chronic stroke reduced upper limb motor functions compared with the period prior to interruption, FMA-UE Eqs (1) and (2) scores were compared using a survey period-based generalized linear model (GLM). A linear model and a Gaussian distribution were used for the GLM. Akaike's information criterion (AIC) was used to test model fit. Age, sex, severity of FMA-UE before interruption, post-onset period (months), and suspended period (days) were used as covariates. For survey items that showed significant model fit, the 95% confidence interval of minimum detectable change (MDC$_{95}$) was calculated using the values at 6 and 3 months prior, and changes were compared. MDC$_{95}$ and standard error of measurement (SEM) were calculated using the following equations:

$$\text{MDC}_{95} = \text{SEM} \times 1.96 \times \sqrt{2}, \tag{3}$$

$$\text{SEM} = \text{SD} \times \sqrt{(1 - \text{ICC})}. \tag{4}$$

A proportion test was used to analyze the questionnaire (bivariate) in chronic stroke patients with interrupted outpatient rehabilitation due to the spread of SARS-CoV-2 infection. To test the hypothesis that symptoms and difficulty in ADL occurred, mental and physical functions were compared with the those before interruption. For statistical analysis, JAMOVI version 1.6.4 (JAMOVI Project, Sydney, Australia) was used, and the statistical significance level was set at 5%.

## Ethical considerations

All patients provided written consent to participate in this study. This study was approved by the Jikei University School of Medicine Ethics Committee (approval number 24-295-7061).

The patients were examined by a doctor before the survey. Patients with fever with a body temperature ($\geq 37°C$), upper respiratory tract inflammation, malaise, taste or olfactory symptoms, or other cold symptoms at the time of examination were prohibited from proceeding. All patients washed their hands and sterilized with alcohol before entering the rehabilitation room to prevent transmission of infection to investigators and patients. Masks were kept donned during the measurement of functional evaluation. Occupational therapists wore masks, face guards, gowns, and rubber gloves while they were inside the rehabilitation room.

## Results

Eighty-one patients met the eligibility criteria between June 1, 2019, and May 31, 2020. Of the 81 patients, 4 patients were younger than 20 years, 4 had completed occupational therapy intervention before outpatient rehabilitation was interrupted due to spread of SARS-CoV-2 infection, 21 did not wish to resume outpatient occupational therapy, and 3 did not provide consent, and the final number of patients to be analyzed was 49, accounting for 60% of the participants surveyed (Fig 2). Table 2 shows the patient characteristics.

The FMA-UE total (main effect of period; $\chi^2$ = 5.68, p = 0.02, AIC = 527) and part A scores ($\chi^2$ = 4.84, p = 0.03, AIC = 452) were significantly model fitted by survey period as results of GLM (Tables 3 and 4). As estimated by GLM, the FMA-UE scores decreased by 1.64 points (z = −2.38, p = 0.02) for the total score and by 1.04 points (z = −2.20, p = 0.03) for part A score due to the 3-month interruption in outpatient rehabilitation (Fig 3). GLM was not fitted to FMA-UE part B ($\chi^2$ = 2.48, p = 0.12, AIC = 325), FMA-UE part C ($\chi^2$ = 0.04, p = 0.85, AIC = 337), FMA-UE part D ($\chi^2$ = 0.71, p = 0.40, AIC = 286), ARAT grasp ($\chi^2$ = 0.45, p = 0.50,

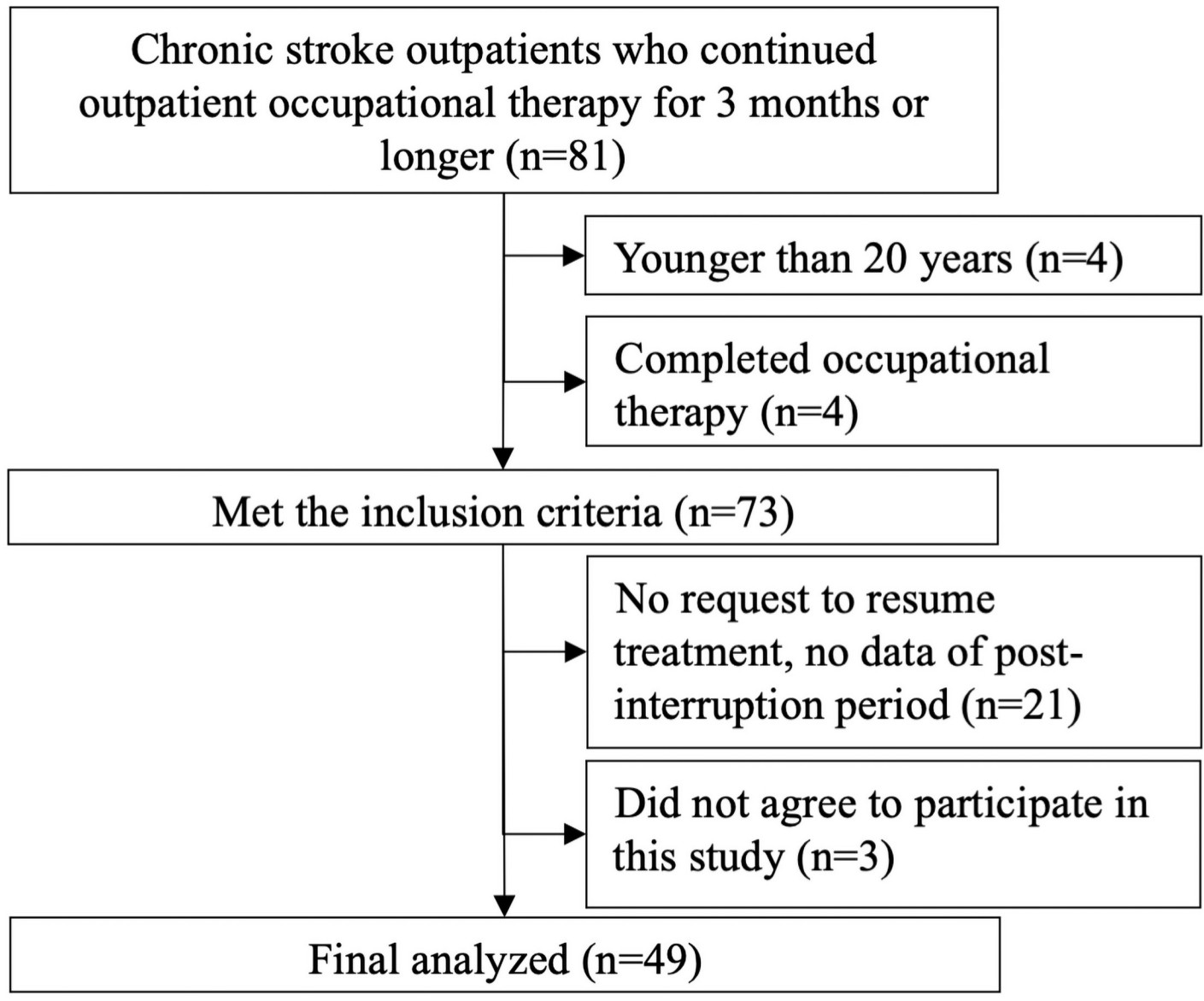

**Fig 2. Acquisition procedure of patients.**

AIC = 378), ARAT grip ($\chi^2$ = 1.53, p = 0.22, AIC = 304), ARAT pinch ($\chi^2$ = 0.95, p = 0.33, AIC = 339), ARAT gross movement ($\chi^2$ = 0.21, p = 0.65, AIC = 172), ARAT total score ($\chi^2$ = 0.07, p = 0.79, AIC = 520), or BI ($\chi^2$ = 0.95, p = 0.33, AIC = 418). The calculated $MDC_{95}$ was 3.58 for FMA-UE part A and 4.50 for FMA-UE total. There was a significant bias toward "no" answers in the proportion test, the first question ("Do you feel that your sleep is getting disturbed?"), and the second question ("Did you get more pain somewhere in your body?") (p<0.05). There was no bias in the distribution of answers to questions 3 ("Do you feel that your joints have become difficult to move?"), 4 ("Do you feel that your muscles are weakened?"), 5 ("Do you feel your muscle tone has increased?"), and 6 ("Do you find your arms and hands difficult to move?"). All 16 questions regarding activities and participation items were significantly biased toward "no" answers (p<0.05) (Table 5).

**Table 2. Characteristics of analyzed patients.**

| Characteristics | Female | Male | All |
|---|---|---|---|
| Participants | 22 (45) | 27 (55) | 49 (100) |
| Age (years) | 50 [42, 62] | 52 [49, 59] | 51 [47, 59] |
| Height (cm) | 157 [153, 164] | 170 [165, 173] | 165 [158, 170] |
| Weight (kg) | 53 [50, 61] | 67 [61, 74] | 62 [53, 70] |
| Paralysis side | | | |
| Left | 7 (32) | 8 (30) | 15 (31) |
| Right | 15 (68) | 19 (70) | 34 (69) |
| Dominant hand | | | |
| Left | 0 (0) | 0 (0) | 0 (0) |
| Right | 22 (100) | 27 (100) | 49 (100) |
| Diagnosis | | | |
| CI | 9 (41) | 11 (41) | 20 (41) |
| ICH | 13 (59) | 16 (59) | 29 (59) |
| Time from onset (months) | 140 [94, 210] | 96 [70, 140] | 116 [81, 155] |
| Treatment by botulinum neurotoxins | | | |
| Treatment | 22 (100) | 27 (100) | 49 (100) |
| No treatment | 0 (0) | 0 (0) | 0 (0) |
| Interruption period (months) | 3 [3] | 3 [3] | 3 [3] |
| Period from 6 months to start date of formal interruption (months) | 7 [5, 9] | 6 [5, 7] | 6 [5, 8] |
| Period from 3 months to start date of formal interruption (months) | 3 [2, 5] | 2 [2, 4] | 3 [2, 4] |
| FMA-UE severity | | | |
| Severe (total score ≤19) | 3 (14) | 4 (15) | 7 (14) |
| Moderate (20< total score <46) | 13 (59) | 21 (78) | 34 (69) |
| Mild (total score ≥47) | 6 (27) | 2 (7) | 8 (16) |

Values are n (%) or median [25th, 75th percentile]. CI, cerebral infarction; ICH, intracranial hemorrhage; FMA-UE, Fugl-Meyer Assessment of the Upper Extremity.

## Discussion

The results of the present study suggest that the FMA-UE decreased by approximately 1.6 points due to inactivity associated with the COVID-19 pandemic within a 3-month interruption of rehabilitation. We also tested whether these interruption of outpatient rehabilitation and prolonged home stay caused symptoms related to mental and physical functions and difficulty in ADL compared with survey periods before the interruption in outpatients with chronic stroke.

The proportion test with bivariate answers to the questionnaire given after outpatient rehabilitation was resumed showed no bias in the distribution of answers to four questions regarding difficulty in moving joints, weakening of muscles, increased muscle tone, and difficulty in moving arms and hands. This suggests that half of the outpatients had subjective difficulty regarding upper limb paralysis in this study. Two questions related to sleep disturbance and physical pain in mental and physical function items as well as all 16 questions regarding the activity and participation items were significantly biased toward "no" answers, suggesting that subjective ADL did not change even if outpatient rehabilitation was interrupted and patients refrained from going out.

The results of this study indicated that the FMA-UE score was significantly reduced, and patients complained of subjective symptoms related to mental and physical functions due to

**Table 3. Generalized linear model.**

| Index of measurements | Score, median [25th, 75th percentile] | | | Estimated marginal means (95% CI lower, upper) | | Log-likelihood ratio tests | | | |
|---|---|---|---|---|---|---|---|---|---|
| | Prior 6 m | Pre IP | Post IP | ΔPre-interruption | ΔPost-interruption | Comparison | SE | Z | p |
| **FMA-UE** | | | | | | | | | |
| A | 25 [20, 29] | 25 [20, 29] | 24 [20, 27] | 0.04 (−0.62, 0.69) | −1.00 (−1.65, −0.34) | −1.04 (−1.96, −0.11) | 0.47 | −2.20 | 0.03 |
| B | 3 [1, 6] | 4 [1, 6] | 2 [1, 5] | 0.12 (−0.22, 0.46) | −0.27 (−0.61, 0.08) | −0.39 (−0.87, 0.10) | 0.25 | −1.57 | 0.12 |
| C | 2 [2, 6] | 2 [2, 5] | 2 [2, 4] | −0.24 (−0.61, 0.12) | −0.29 (−0.66, 0.07) | −0.05 (−0.56, 0.46) | 0.26 | −0.19 | 0.85 |
| D | 0 [0, 3] | 0 [0, 3] | 0 [0, 2] | −0.02 (−0.30, 0.26) | −0.19 (−0.47, 0.09) | −0.17 (−0.57, 0.23) | 0.20 | −0.85 | 0.40 |
| Total | 33 [24, 39] | 31 [24, 42] | 31 [23, 38] | −0.10 (−1.07, 0.86) | −1.75 (−2.71, −0.79) | −1.64 (−3.00, −0.29) | 0.69 | −2.38 | 0.02 |
| **ARAT** | | | | | | | | | |
| Grasp | 0 [0, 4] | 0 [0, 4] | 0 [0, 2] | −0.13 (−0.58, 0.32) | −0.34 (−0.79, 0.11) | −0.22 (−0.85, 0.42) | 0.32 | −0.67 | 0.51 |
| Grip | 0 [0, 4] | 0 [0, 4] | 0 [0, 3] | −0.12 (−0.43, 0.19) | −0.39 (−0.70, −0.08) | −0.27 (−0.71, 0.16) | 0.22 | −1.24 | 0.22 |
| Pinch | 0 [0, 0] | 0 [0, 0] | 0 [0, 0] | −0.34 (−0.71, 0.03) | −0.08 (−0.45, 0.29) | 0.26 (−0.26, 0.78) | 0.26 | 0.98 | 0.33 |
| Gross movement | 4 [3, 5] | 3 [3, 5] | 3 [3, 5] | −0.17 (−0.32, −0.01) | −0.11 (−0.27, 0.04) | 0.05 (−0.17, 0.27) | 0.11 | 0.46 | 0.65 |
| Total | 4 [3, 12] | 4 [3, 12] | 3 [3, 8] | −0.75 (−1.68, 0.18) | −0.93 (−1.86, −0.01) | −0.18 (−1.49, 1.13) | 0.68 | −0.27 | 0.79 |
| **BI** | 100 [100] | 100 [100] | 100 [100] | −0.08 (−0.62, 0.47) | −0.46 (−1.01, 0.09) | −0.38 (−1.16, 0.39) | 0.39 | −0.97 | 0.33 |

Data were retrieved at 6 and 3 months before interruption of outpatient rehabilitation. CI, confidence interval; Pre 6 m, approximately 6 months before outpatient rehabilitation was interrupted; Pre IP, approximately 3 months before outpatient rehabilitation was interrupted; Post IP, resumption of outpatient rehabilitation; ΔPre-interruption, changes during the 3-month period before the interruption of rehabilitation; ΔPost-interruption, changes during the 3-month period after the resumption of rehabilitation; SE, standard error; FMA-UE, Fugl-Meyer Assessment of the Upper Extremity; ARAT, Action Research Test; BI, Barthel Index. Generalized linear model was used to compare FMA-UE score changes, with statistical significance set at 0.05 (n = 49).

inactivity associated with interrupted rehabilitation and refrained from going out. Stroke patients who lived in their own home or residing in a nursing or residential home often maintain and improve their body function and ADL through community rehabilitation [24]. However, it was also suggested that for patients requiring treatment in the hospital, a decrease in upper limb motor function is a risk factor when outpatient rehabilitation is interrupted and patients had prolonged home stay due to the spread of SARS-CoV-2 infection. FMA is an evaluation battery associated with body function and body structure domains according to the ICF category [25]. The FMA-UE of part A score was reduced due to interruption of outpatient rehabilitation and refraining from going out for approximately 3 months in stroke outpatients; thus, voluntary movement declined and the range of motion in the shoulder, elbow, and forearm were restricted. Meanwhile, outpatients' ADL was maintained, and they rarely complained of difficulties with ADL in this study. The ARAT is associated with activity in the ICF category and has been shown to correlate with scores in the Motor Activity Log, which assesses ADL [25, 26]. BI and ARAT were not significantly reduced due to interruptions in outpatient rehabilitation and the patients refrained from going out in this study. Therefore, hemiplegic

**Table 4. Comparisons of FMA-UE score changes due to interruption periods.**

| FMA-UE | ΔPre-interruption | ΔPost-interruption | Mean difference | SE | df | T | pTukey |
|---|---|---|---|---|---|---|---|
| Total | −0.10 (−1.07, 0.86) | −1.75 (−2.71, −0.79) | 1.64 | 0.69 | 91 | 2.39 | 0.02 |
| Part A | 0.04 (−0.62, 0.69) | −1.00 (−1.65, −0.34) | 1.04 | 0.47 | 91 | 2.20 | 0.03 |

Tukey's test was used for multiple comparisons. Values are Estimated Marginal Means (95%CI lower, upper) (n = 49). FMA-UE, Fugl-Meyer Assessment of the Upper Extremity. ΔPre-interruption, changes during the 3-month period before the interruption of rehabilitation; ΔPost-interruption, changes during the 3-month period after the resumption of rehabilitation; SE, standard error.

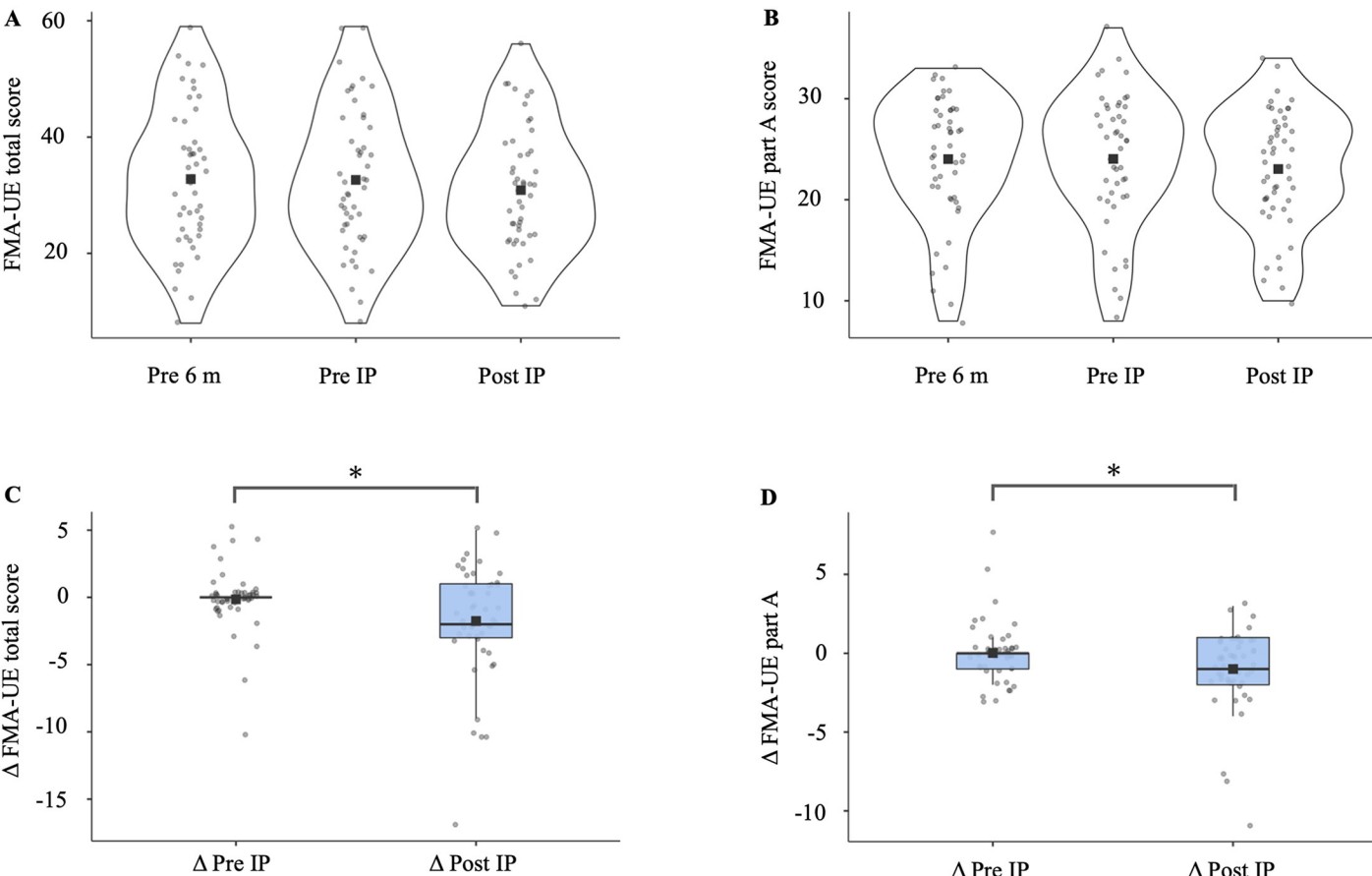

**Fig 3. Comparisons of FMA-UE score changes due to interruption.** (A) FMA-UE total violin plots are shown for each period. (B) The FMA-UE part A values were plotted for each period. The black squares in the graph indicate the mean. The gray circles indicate the values for each patient. (C) Box plots comparing the Δ values of the FMA-UE total (Tukey's test, *p<0.05). (D) Comparison of the Δ values of the FMA-UE part A (Tukey's test, *p<0.05). Statistical significance was set at p<0.05 (n = 49). FMA-UE, Fugl-Meyer Assessment of the Upper Extremity; Pre 6 m, approximately 6 months before outpatient rehabilitation was interrupted; Pre IP, approximately 3 months before outpatient rehabilitation was interrupted; Post IP, resumption of outpatient rehabilitation.

patients need to practice range of motion exercises for the shoulder, elbow, and forearm in order to stretch the muscles involved in these joints and thus prevent the loss of upper limb motor functions due to inactivity associated with the spread of COVID-19. Therapists should specifically monitor patients for changes in motion, range of motion, and muscle tone related to the shoulder, elbow, and forearm and provide appropriate exercise instructions.

Telework and telerehabilitation programs have an important role in stroke patients during the COVID-19 pandemic [27]. The declining birthrate and the aging population will maximize the difference between deaths and births in 2040, and this imbalance is projected to make the situation worse for 80 years, unless it accepts immigrants in Japan. In that respect, the quality of remote rehabilitation is important for maintaining patient motivation and ability. Even if the patient is unable to go to the hospital, with telecommunication technology, remote rehabilitation is an effective means for the therapist to assist patients in practicing properly at home. One of the challenges of telerehabilitation is disease risk management. There is no system in place to deal with changes in medical conditions of patients at home in Japan. In home care, there are issues to be solved, such as how to play the role of doctors and what to do with the patient–doctor–therapist relationship. In addition, telerehabilitation is not yet covered by insurance. There are enormous barriers to the approval of telerehabilitation as an insurance

Table 5. Patient's feelings about his/her physical and life state.

| Questions | Counts (yes) | Proportion | 95% Confidence interval | | p |
|---|---|---|---|---|---|
| | | | Lower | Upper | |
| **Body functions** | | | | | |
| Q1 | 6 | 0.12 | 0.25 | 0.05 | <0.00 |
| Q2 | 7 | 0.14 | 0.27 | 0.06 | <0.00 |
| Q3 | 26 | 0.53 | 0.68 | 0.38 | 0.78 |
| Q4 | 19 | 0.39 | 0.54 | 0.25 | 0.15 |
| Q5 | 29 | 0.59 | 0.73 | 0.44 | 0.25 |
| Q6 | 23 | 0.47 | 0.62 | 0.33 | 0.78 |
| **Activities and participation** | | | | | |
| Q1 | 4 | 0.08 | 0.20 | 0.02 | <0.00 |
| Q2 | 4 | 0.08 | 0.20 | 0.02 | 0.01 |
| Q3 | 5 | 0.10 | 0.22 | 0.03 | <0.00 |
| Q4 | 16 | 0.33 | 0.48 | 0.20 | 0.02 |
| Q5 | 10 | 0.20 | 0.34 | 0.10 | <0.00 |
| Q6 | 15 | 0.31 | 0.45 | 0.18 | 0.01 |
| Q7 | 3 | 0.06 | 0.17 | 0.01 | <0.00 |
| Q8 | 2 | 0.04 | 0.14 | 0.01 | <0.00 |
| Q9 | 4 | 0.08 | 0.20 | 0.02 | <0.00 |
| Q10 | 3 | 0.06 | 0.17 | 0.01 | <0.00 |
| Q11 | 2 | 0.04 | 0.14 | 0.01 | <0.00 |
| Q12 | 12 | 0.25 | 0.39 | 0.13 | <0.00 |
| Q13 | 3 | 0.06 | 0.17 | 0.01 | <0.00 |
| Q14 | 4 | 0.08 | 0.20 | 0.02 | <0.00 |
| Q15 | 10 | 0.20 | 0.34 | 0.10 | <0.00 |
| Q16 | 9 | 0.18 | 0.32 | 0.09 | <0.00 |

A proportion test with binomial responses was used. Statistical significance was set at 0.05 (n = 49).

practice in Japan. As stroke patients are unable to receive outpatient care during pandemic, telerehabilitation is important; thus, the Jikei University School of Medicine is scheduled to start telerehabilitation in the near future.

This study had several limitations. First, the amount of exercise was not measured with activity metrics or pedometers. Because of the COVID-19 epidemic, the government of Japan requested citizens to refrain from outdoor activities. Therefore, it was speculated that patients who used to go out had lower physical activity. Disuse-dependent plasticity in the brain was one of the causes of exacerbation of paralysis. In addition, whether interruption of rehabilitation and restriction in outdoor activities have affected patients with stroke in this study in unknown. In our next study, it is necessary to measure and investigate the degree of activity reduction caused by upper limb motor paralysis using activity metrics. Second, the effects of botulinum toxin type A (BoNT-A) injections were not considered in this study. All patients in this study received BoNT-A treatment. BoNT-A is an effective treatment for reducing spasticity in patients with stroke [28, 29]. It has also been reported to be effective in improving insomnia and relieving pain for patients who have continued treatment for a long period [30]. It is of concern that decreased upper limb motor function and subjective symptoms may occur in patients who could not be injected with BoNT-A, as the effect of BoNT-A is maintained for only approximately 3 months after injection [31, 32]. It was unclear whether upper limb motor function declined and subjective symptoms occurred in patients who did not receive BoNT-A

injection during the rehabilitation discontinuation period. Third, most of the patients in this study had high ADLs and were generally self-reliant. Another study should clarify the situation in which physical function and ADL are impaired in critically ill chronic patients who require assistance with ADL. Fourth, the questionnaire survey conducted after rehabilitation resumed did not examine the reproducibility of the results. Fifth, this study was conducted at a single university hospital in Tokyo. It is unclear whether the same results as those in this study were obtained in other areas of Japan or facilities with other hospital functions. The results of larger studies should be referred to for the effects of out-of-home and behavioral restrictions imposed by untreatable pathogens on physical function.

## Conclusion

The results of this study suggest that upper limb motor paralysis was slightly exacerbated in patients with chronic stroke whose activity was restricted due to COVID-19 pandemic. This result can be used as a reference value for the amount of upper limb dysfunction that occurs in stroke patients when outpatient rehabilitation is unavailable and patients refrain from going out for approximately 3 months. Further research is needed to determine whether upper limb motor function recovery can be measured when treatment is resumed in these patients.

## Acknowledgments

We would like to thank all occupational therapists of the Department of Rehabilitation, Jikei University School of Medicine, for their cooperation in obtaining operational approval and data for conducting this study.

## Author Contributions

**Conceptualization:** Toyohiro Hamaguchi, Masahiro Abo.

**Data curation:** Daigo Sakamoto, Yasuhide Nakayama, Takuya Hada.

**Formal analysis:** Daigo Sakamoto, Toyohiro Hamaguchi.

**Investigation:** Yasuhide Nakayama, Takuya Hada.

**Methodology:** Daigo Sakamoto, Toyohiro Hamaguchi, Masahiro Abo.

**Supervision:** Toyohiro Hamaguchi, Yasuhide Nakayama, Takuya Hada, Masahiro Abo.

**Validation:** Toyohiro Hamaguchi.

**Visualization:** Toyohiro Hamaguchi.

**Writing – original draft:** Daigo Sakamoto.

**Writing – review & editing:** Toyohiro Hamaguchi, Yasuhide Nakayama, Takuya Hada, Masahiro Abo.

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
