## [Decision Letter · Decision Letter 0]

24 Aug 2021

PONE-D-21-12560

Changes in motor paralysis involving upper extremities of outpatient chronic stroke patients from temporary rehabilitation interruption due to spread of COVID-19 infection: An observational study on pre and post survey data without a control group

PLOS ONE

Dear Dr. Hamaguchi,

Thank you for submitting your manuscript to PLOS ONE. After careful consideration, we feel that it has merit but does not fully meet PLOS ONE’s publication criteria as it currently stands. Therefore, we invite you to submit a revised version of the manuscript that addresses the points raised during the review process.

There are clarifications requested by both reviewers on methods and results collections as well as more precise and extended mention to previous work in the intro and message for practice in the discussion. 

We look forward to receiving your revised manuscript.

Kind regards,

Andrea Martinuzzi

Academic Editor

PLOS ONE

Journal Requirements:

Reviewers' comments:

Reviewer's Responses to Questions

**Comments to the Author**

1. Is the manuscript technically sound, and do the data support the conclusions?

Reviewer #1: Yes

Reviewer #2: Partly

2. Has the statistical analysis been performed appropriately and rigorously? 

Reviewer #1: Yes

Reviewer #2: I Don't Know

3. Have the authors made all data underlying the findings in their manuscript fully available?

Reviewer #1: Yes

Reviewer #2: Yes

4. Is the manuscript presented in an intelligible fashion and written in standard English?

Reviewer #1: Yes

Reviewer #2: Yes

5. Review Comments to the Author

Reviewer #1: Dear authors: Thank you for an interesting paper.

- Abstract:

o Concise and readable.

- Introduction:

o It would be nice if the authors more deeply refer to previous studies on the importance of telerehabilitation and telework in stroke patients during unpredictable situation such as COVID-19. In this case, the authors can find the important aspects of telerehabilitation of patients with a stroke in the following commentary article: “Telework and telerehabilitation programs for workers with a stroke during the COVID-19 pandemic: A commentary”.

- Methods

o Sample size was calculated using G*power software with power set at 0.80, type I error rate of 0.05, and an effect size 0.05. Please clarify that whether this was based on priori analysis or post-hoc test.

o Line 119: The full name of the “Action Research Arm Test” and “the Barthel Index” should be presented before use of abbreviation.

- Results:

o Table 2: In participant’s section, (n) shows the number of participants. But the percentages of data are reported. Please clarified this section.

- Discussion:

o Excellent.

- Conclusion:

o Relevant to the main findings.

Reviewer #2: The manuscript reports on the possible effects of a) interruption of outpatient rehabilitation and b) limitation in outdoor activities due to COVID19 epidemic in a population of adults affected by long-term sequelae of stroke. The authors focus on upper limb function.

The paper addresses an interesting aspect, such as the impact of the pandemic on the delivery of rehabilitation intervention; however, the possible different role of the two factors reported (interruption of rehabilitation vs. restriction in outdoor activities) is not clarified.

The results suggest that the overall effect of the above-mentioned factors on upper limb function is relatively small, and essentially limited to the intrinsic upper limb motricity, with no detectable impact on the ADL.

The authors’ conclusions are that therapists should monitor the upper limb motricity in patients with long term stroke who suspend outpatient treatment and (only mentioned in the abstract) give advice on self-training activities.

The methods and instruments utilized in the survey protocol are adequate for the specific aspects investigated in the study.

The authors should clarify the procedures of administration of the ad hoc questionnaire (has it been administered comparing PreIP vs PostIP and Pre6m vs PreIP? oj just PreIP vs PostIP?).

The statistical analysis seems to be appropriate; however, I must declare not to have enough expertise to make a sound judgment.

Some limitations of the study are adequately reported in the discussion; as for the possible effect of Botulinum injection, as 100% of the sample underwent the procedure, one possible explanation of the decrease in upper limb motricity could exactly be attributed to the gradual decrease of the effect of the drug.

Some points in the discussion seem to repeat information already given in the results (e.g. the decrease in score of FMA-UE total and part A).

The possible impact of the results on the therapists’ practice could be discussed in more detail (e.g. the issue of self-training is pointed out in the abstract only).

As a minor remark, the ICF code for sleep disturbances (table 1) is b134 instead of b139

6. PLOS authors have the option to publish the peer review history of their article (what does this mean?). If published, this will include your full peer review and any attached files.

Reviewer #1: **Yes: **Taher Babaee

Reviewer #2: **Yes: **PAOLO BOLDRINI

---

## [Author Response · Author response to Decision Letter 0]

1 Sep 2021

Reviewer #1

Comment 1- Introduction

It would be nice if the authors more deeply refer to previous studies on the importance of telerehabilitation and telework in stroke patients during unpredictable situation such as COVID-19. In this case, the authors can find the important aspects of telerehabilitation of patients with a stroke in the following commentary article: “Telework and telerehabilitation programs for workers with a stroke during the COVID-19 pandemic: A commentary”.

Response: We thank Reviewer #1 for this comment. As suggested, we added some text to the Discussion section based on the commentary by Moradi et al. [“Telework and telerehabilitation programs for workers with a stroke during the COVID-19 pandemic: A commentary,” Work. 2021;68(1): 77-80. doi: 10.3233/WOR-203356]:

“Telework and telerehabilitation programs have an important role in stroke patients during the COVID-19 pandemic [27]. The declining birthrate and the aging population will maximize the difference between deaths and births in 2040, and this imbalance is projected to make the situation worse for 80 years, unless it accepts immigrants in Japan. In that respect, the quality of remote rehabilitation is important for maintaining patient motivation and ability. Even if the patient is unable to go to the hospital, with telecommunication technology, remote rehabilitation is an effective means for the therapist to assist patients in practicing properly at home. One of the challenges of telerehabilitation is disease risk management. There is no system in place to deal with changes in medical conditions of patients at home in Japan. In home care, there are issues to be solved, such as how to play the role of doctors and what to do with the patient–doctor–therapist relationship. In addition, telerehabilitation is not yet covered by insurance. There are enormous barriers to the approval of telerehabilitation as an insurance practice in Japan. As stroke patients are unable to receive outpatient care during pandemic, telerehabilitation is important; thus, the Jikei University School of Medicine is scheduled to start telerehabilitation in the near future.” (Page 30, lines 286-297, to page 31, lines 298-300)

Comment 2- Methods

Sample size was calculated using G*power software with power set at 0.80, type I error rate of 0.05, and an effect size 0.05. Please clarify that whether this was based on priori analysis or post-hoc test.

Response: We initially calculated the sample size power analysis based on prior analysis. According to the comments, the text has been revised as follows:

“Changes in the FMA-UE total score was used to fit the linear model. Sample size was estimated by a priori analysis using G*Power, with χ2 tests with goodness-of-fit set at α = 0.05, 1-β= 0.8, and effect size of 0.5 [18].” (Page 6-7, lines 94-96)

Comment 3- Methods

Line 119: The full name of the “Action Research Arm Test” and “the Barthel Index” should be presented before use of abbreviation.

Response: We have added the abbreviations of “Action Research Arm Test” and “the Barthel Index” as follows:

“The data for the Pre 6 m and Pre IP periods were surveyed retrospectively from medical records using FMA-UE, Action Research Arm Test (ARAT), and Barthel Index (BI) data.” (Page 8, lines 113-115)

Comment 4- Results

Table 2: In participant’s section, (n) shows the number of participants. But the percentages of data are reported. Please clarified this section.

Response: Table 2 did not match the text in our explanation of the results regarding patients’ recruitment; hence, we added to the following:

“the final number of patients to be analyzed was 49, accounting for 60% of the participants surveyed.” (Page 15, lines 208-209)

We also changed the title of Table 2 to “Characteristics of Analyzed Patients.” (Page 17)

Reviewer #2

Comment 1

The manuscript reports on the possible effects of a) interruption of outpatient rehabilitation and b) limitation in outdoor activities due to COVID19 epidemic in a population of adults affected by long-term sequelae of stroke. The authors focus on upper limb function.

The paper addresses an interesting aspect, such as the impact of the pandemic on the delivery of rehabilitation intervention; however, the possible different role of the two factors reported (interruption of rehabilitation vs. restriction in outdoor activities) is not clarified.

Response: We thank Reviewer #2 for thoroughly understanding our manuscript. In this study, it was not possible to clarify which of the possible different roles of the two factors, interruption of rehabilitation and restriction in outdoor activities, caused exacerbations of motor paralysis in chronic patients with stroke. Therefore, we added text in the Discussion section as follows:

“Telework and telerehabilitation programs have an important role in stroke patients during the COVID-19 pandemic [27]. The declining birthrate and the aging population will maximize the difference between deaths and births in 2040, and this imbalance is projected to make the situation worse for 80 years, unless it accepts immigrants in Japan. In that respect, the quality of remote rehabilitation is important for maintaining patient motivation and ability. Even if the patient is unable to go to the hospital, with telecommunication technology, remote rehabilitation is an effective means for the therapist to assist patients in practicing properly at home. One of the challenges of telerehabilitation is disease risk management. There is no system in place to deal with changes in medical conditions of patients at home in Japan. In home care, there are issues to be solved, such as how to play the role of doctors and what to do with the patient–doctor–therapist relationship. In addition, telerehabilitation is not yet covered by insurance. There are enormous barriers to the approval of telerehabilitation as an insurance practice in Japan. As stroke patients are unable to receive outpatient care during pandemic, telerehabilitation is important; thus, the Jikei University School of Medicine is scheduled to start telerehabilitation in the near future.” (Page 28, lines 286-295, to page 29, lines 296-300)

Comment 2

The results suggest that the overall effect of the above-mentioned factors on upper limb function is relatively small, and essentially limited to the intrinsic upper limb motricity, with no detectable impact on the ADL. The authors’ conclusions are that therapists should monitor the upper limb motricity in patients with long term stroke who suspend outpatient treatment and (only mentioned in the abstract) give advice on self-training activities. The methods and instruments utilized in the survey protocol are adequate for the specific aspects investigated in the study.　The authors should clarify the procedures of administration of the ad hoc questionnaire (has it been administered comparing PreIP vs PostIP and Pre6m vs PreIP? or just PreIP vs PostIP?). The statistical analysis seems to be appropriate; however, I must declare not to have enough expertise to make a sound judgment.

Response: The ad hoc questionnaire was given to each patient only once. As for the statistics of the questionnaire, the distribution of yes/no was examined. The results of this questionnaire were not compared pre- and post-IP. To clarify this, we modified the text as follows:

“To investigate any functional changes that occurred during interruptions of outpatient rehabilitation, each patient answered a questionnaire once after resuming outpatient rehabilitation.” (Page 8, lines 113-115)

Comment 3

Some limitations of the study are adequately reported in the discussion; as for the possible effect of Botulinum injection, as 100% of the sample underwent the procedure, one possible explanation of the decrease in upper limb motricity could exactly be attributed to the gradual decrease of the effect of the drug.

Response: We agree with this comment and rewrote the text in second limitation as follows:

“Second, the effects of botulinum toxin type A (BoNT-A) injections were not considered in this study. All patients in this study received BoNT-A treatment. BoNT-A is an effective treatment for reducing spasticity in patients with stroke [28,29]. It has also been reported to be effective in improving insomnia and relieving pain for patients who have continued treatment for a long period [30]. It is of concern that decreased upper limb motor function and subjective symptoms may occur in patients who could not be injected with BoNT-A, as the effect of BoNT-A is maintained for only approximately 3 months after injection [31,32]. It was unclear whether upper limb motor function declined and subjective symptoms occurred in patients who did not receive BoNT-A injection during the rehabilitation discontinuation period.” (Page 29, lines 308-311, to page 30, lines 312-317)

Comment 4

Some points in the discussion seem to repeat information already given in the results (e.g. the decrease in score of FMA-UE total and part A).

Response: We agree with this comment. We removed the duplicate from the first paragraph of the Discussion section and modified the text as follows:

“The results of the present study suggest that the FMA-UE decreased by approximately 1.6 points due to inactivity associated with the spread of COVID-19 infection within a 3-month interruption of rehabilitation. We also tested whether interruption of outpatient rehabilitation and prolonged home stay caused symptoms related to mental and physical functions and difficulty in ADL compared with survey periods before the interruption in outpatients with chronic stroke.” (Page 26, lines 249-254)

Comment 5

The possible impact of the results on the therapists’ practice could be discussed in more detail (e.g. the issue of self-training is pointed out in the abstract only).

Response: We discussed the possible impact and added the following to the Discussion section:

“Even if the patient is unable to go to the hospital, with telecommunication technology, remote rehabilitation is an effective means for the therapist to assist patients in practicing properly at home. One of the challenges of telerehabilitation is disease risk management. There is no system in place to deal with changes in medical conditions of patients at home in Japan. In home care, there are issues to be solved, such as how to play the role of doctors and what to do with the patient–doctor–therapist relationship. In addition, telerehabilitation is not yet covered by insurance. There are enormous barriers to the approval of telerehabilitation as an insurance practice in Japan. As stroke patients are unable to receive outpatient care during pandemic, telerehabilitation is important; thus, the Jikei University School of Medicine is scheduled to start telerehabilitation in the near future.” (Page 28, lines 290-295, to page 29, lines 296-300)

Comment 6

As a minor remark, the ICF code for sleep disturbances (table 1) is b134 instead of b139

Response: We have fixed the ICF code in Table 1. In addition, we also changed the code of gait disturbance to d450.

---

## [Decision Letter · Decision Letter 1]

17 Nov 2021

Changes in motor paralysis involving upper extremities of outpatient chronic stroke patients from temporary rehabilitation interruption due to spread of COVID-19 infection: an observational study on pre- and post-survey data without a control group

PONE-D-21-12560R1

Dear Dr. Hamaguchi,

We’re pleased to inform you that your manuscript has been judged scientifically suitable for publication and will be formally accepted for publication once it meets all outstanding technical requirements.

Kind regards,

Andrea Martinuzzi

Academic Editor

PLOS ONE

Additional Editor Comments (optional):

Reviewers' comments:

Reviewer's Responses to Questions

**Comments to the Author**

1. If the authors have adequately addressed your comments raised in a previous round of review and you feel that this manuscript is now acceptable for publication, you may indicate that here to bypass the “Comments to the Author” section, enter your conflict of interest statement in the “Confidential to Editor” section, and submit your "Accept" recommendation.

Reviewer #1: All comments have been addressed

2. Is the manuscript technically sound, and do the data support the conclusions?

Reviewer #1: Yes

3. Has the statistical analysis been performed appropriately and rigorously? 

Reviewer #1: Yes

4. Have the authors made all data underlying the findings in their manuscript fully available?

Reviewer #1: Yes

5. Is the manuscript presented in an intelligible fashion and written in standard English?

Reviewer #1: Yes

6. Review Comments to the Author

Reviewer #1: Thank you for your thorough revision of the present form of above-mentioned manuscript.

All the queries have addressed.

7. PLOS authors have the option to publish the peer review history of their article (what does this mean?). If published, this will include your full peer review and any attached files.

Reviewer #1: **Yes: **Taher Babaee

---

## [Editor Report · Acceptance letter]

1 Dec 2021

PONE-D-21-12560R1 

Changes in motor paralysis involving upper extremities of outpatient chronic stroke patients from temporary rehabilitation interruption due to spread of COVID-19 infection: an observational study on pre- and post-survey data without a control group 

Dear Dr. Hamaguchi:

I'm pleased to inform you that your manuscript has been deemed suitable for publication in PLOS ONE. Congratulations! Your manuscript is now with our production department. 

Kind regards, 

on behalf of

Dr. Andrea Martinuzzi 

Academic Editor

PLOS ONE